# Novel Selective PPARα Modulator Pemafibrate for Dyslipidemia, Nonalcoholic Fatty Liver Disease (NAFLD), and Atherosclerosis [note 1]

**DOI:** 10.3390/metabo13050626

**Published:** 2023-05-02

**Authors:** Shizuya Yamashita, Manfredi Rizzo, Ta-Chen Su, Daisaku Masuda

**Affiliations:** 1Department of Cardiology, Rinku General Medical Center, Izumisano 598-8577, Osaka, Japan; 2Department of Internal Medicine and Medical Specialties, School of Medicine, University of Palermo, 90133 Palermo, Italy; 3Promise Department, School of Medicine, University of Palermo, 90133 Palermo, Italy; 4Department of Environmental and Occupational Medicine, National Taiwan University Hospital, Taipei 10002, Taiwan; 5Institute of Environmental and Occupational Health Sciences, College of Public Health, National Taiwan University, Taipei 10017, Taiwan; 6Division of Cardiology, Department of Internal Medicine, National Taiwan University Hospital, Taipei 10002, Taiwan

**Keywords:** PPARα, SPPARMα, pemafibrate, triglycerides, dyslipidemia

## Abstract

Statins, the intestinal cholesterol transporter inhibitor (ezetimibe), and PCSK9 inhibitors can reduce serum LDL-C levels, leading to a significant reduction in cardiovascular events. However, these events cannot be fully prevented even when maintaining very low LDL-C levels. Hypertriglyceridemia and reduced HDL-C are known as residual risk factors for ASCVD. Hypertriglyceridemia and/or low HDL-C can be treated with fibrates, nicotinic acids, and n-3 polyunsaturated fatty acids. Fibrates were demonstrated to be PPARα agonists and can markedly lower serum TG levels, yet were reported to cause some adverse effects, including an increase in the liver enzyme and creatinine levels. Recent megatrials of fibrates have shown negative findings on the prevention of ASCVD, which were supposed to be due to their low selectivity and potency for binding to PPAR α. To overcome the off-target effects of fibrates, the concept of a selective PPARα modulator (SPPARMα) was proposed. Kowa Company, Ltd. (Tokyo, Japan), has developed pemafibrate (K-877). Compared with fenofibrate, pemafibrate showed more favorable effects on the reduction of TG and an increase in HDL-C. Fibrates worsened liver and kidney function test values, although pemafibrate showed a favorable effect on liver function test values and little effect on serum creatinine levels and eGFR. Minimal drug–drug interactions of pemafibrate with statins were observed. While most of the fibrates are mainly excreted from the kidney, pemafibrate is metabolized in the liver and excreted into the bile. It can be used safely even in patients with CKD, without a significant increase in blood concentration. In the megatrial of pemafibrate, PROMINENT, for dyslipidemic patients with type 2 diabetes, mild-to-moderate hypertriglyceridemia, and low HDL-C and LDL-C levels, the incidence of cardiovascular events did not decrease among those receiving pemafibrate compared to those receiving the placebo; however, the incidence of nonalcoholic fatty liver disease was lower. Pemafibrate may be superior to conventional fibrates and applicable to CKD patients. This current review summarizes the recent findings on pemafibrate.

## 1. Introduction

Currently available fibrates target PPARα, eliciting their biological effects, including the reduction of serum TG levels and an increase in serum HDL-C levels [1,2]. Although fibrates-mediated PPARα activation improves serum TG and HDL-C levels, off-target effects such as abnormal hepatic and renal function tests were often noted. In megatrials of fibrates, such as HHS [3] and VA-HIT [4], gemfibrozil reduced the CV event rate significantly. However, when gemfibrozil was used in combination with cerivastatin, rhabdomyolysis often occurred [5]. Afterward, cerivastatin disappeared from the market.

In subsequent megatrials such as the BIP Study, which used bezafibrate [6], the FIELD Study, which used fenofibrate in type 2 diabetic patients [7], and the ACCORD-Lipid Study, which used fenofibrate added to simvastatin in type 2 diabetic patients [8], all primary endpoints were statistically negative. Thus, the administration of fibrates could not prove clinical benefits to prevent CV events. 

However, when the effects of the fibrates were meta-analyzed [9,10], the CV event rate was significantly reduced. Furthermore, the ACCORD Study was followed up with the ACCORDION Study, which showed a favorable effect of fenofibrate on long-term CV risk [11]. Reduced mortality was also shown in hypertriglyceridemic patients at the baseline in the BIP Study [12]. Later meta-analyses showed supportive evidence of fibrates in both primary and secondary CV prevention [13,14]. In contrast to the Cholesterol Treatment Trialists’ (CTT) Collaboration meta-analyses [15,16,17], a significant decrease in total mortality was not demonstrated by fibrates [9,10,13,14]. Adverse effects of fibrates such as abnormal liver and kidney function test values may have overwhelmed their benefits. Therefore, new treatment options for atherogenic dyslipidemia in patients with metabolic syndrome, obesity, diabetes mellitus, and ASCVD have been long awaited.

## 2. Development of Selective Peroxisome Proliferator-Activated Receptor (PPAR)α Modulators (SPPARMα)

Mechanisms through which fibrates exert hypolipidemic effects were not clear until the discovery of PPARs. Fibrates are one of the agonists of PPARα. PPARα affects lipid and lipoprotein metabolism by regulating the transcriptions of genes involved in the metabolism of TG-rich lipoproteins and HDL [1,18]. PPARα activation induces the production of LPL and apo A-V, while it decreases apo C-III, which inhibits LPL activity. Thus, PPARα agonists enhance TG-rich lipoprotein catabolism, which results in a reduction in serum TG levels [19,20,21,22]. PPARα activation also enhances the gene expressions involved in β-oxidation. An enhanced β-oxidation and increased expression of hepatic ACS may reduce fatty acid levels in the liver [23], leading to decreased production of VLDL particles by the liver. PPARα activation decreases atherogenic small dense LDL particles [24]. 

Moreover, PPARα activation increases the synthesis of HDL by enhancing the gene expressions of apo A-I and A-II, major components of HDL [25]. Enhanced LPL activity accelerates TG-rich lipoproteins catabolism and increases the transfer of phospholipids to HDL [26]. PPARα activation increases the gene expression of ABCA1 and ABCG1, which accelerates the efflux of cholesterol from macrophages [27]. Activation of PPARα increases the expression of SR-BI in the liver, which enhances the selective hepatic uptake of cholesteryl ester via HDL [28]. Activation of PPARα also regulates glucose homeostasis, attenuates inflammation and thrombogenesis, and improves vascular function [29]. Activation of PPARα ameliorates abnormal lipid and glucose metabolism and downregulates the expression of proinflammatory genes in monocytes/macrophages, which may lead to vascular protection against atherothrombosis.

To overcome the negative profiles of fibrates, Fruchart put forward a new concept of SPPARMα [30,31]. This concept is similar to that of SERMs [32]. Tamoxifen is the first estrogen receptor modulator that has antiestrogenic activity in mammary glands and partial proestrogenic activity in bones and the uterus. However, contrary to expectations, the long-term use of tamoxifen enhanced the incidence of uterine cancer. Then, raloxifene, a second-generation SERM with tissue-specific activity was later developed.

Kowa Company, Ltd. (Japan) has screened >1500 compounds to identify SPPARMα, which possess tissue-specific and targeted gene-selective activities. They finally identified several candidates of agonists possessing a very potent PPARα activity as well as high selectivity of PPARα. Kowa identified three compounds (R-24, R-35, and R-36) as candidates, finally selecting R-36, named pemafibrate (K-877, Parmodia^®^ tablet). We previously reviewed the development of pemafibrate for the application of dyslipidemia and atherosclerosis [33,34]. Pemafibrate could activate PPARα > 2500 times more strongly than fenofibric acid (the active form of fenofibrate). Pemafibrate is extremely selective for PPARα (>5000-fold for PPARγ and >11,000-fold for PPARδ, respectively) [35,36]. 

In preclinical studies, pemafibrate reduced serum TG and increased HDL-C levels more strongly than fenofibrate, by inhibiting VLDL secretion and enhancing TG clearance through activating LPL [37]. Moreover, the expression of the VLDL receptor was increased by pemafibrate, leading to enhanced catabolism of VLDL and VLDL remnants [38]. Postprandial hyperlipidemia was attenuated by pemafibrate via inhibition of intestinal cholesterol transporter NPC1L1 mRNA expression in small intestinal mucosa in mice fed a high-fat diet [39,40]. Attenuation of postprandial hyperlipidemia may result from suppression of chylomicron synthesis and secretion by inhibiting NPC1L1-mediated cholesterol absorption as well as PPARα activation in the small intestines. Pemafibrate enhances the expression of genes related to fatty acid β-oxidation and decreases VLDL secretion from the liver [35,39]. FGF21 accelerates fatty acid β-oxidation and PPARα is known to regulate the expression of FGF21 [41,42], which decreases hepatic VLDL secretion by regulating fatty acids uptake by adipose tissue [43]. Pemafibrate increases serum levels and tissue expression of FGF21 [35,39,44]. Upregulation of FGF21 via PPARα by pemafibrate may contribute to decreased serum TG and VLDL levels.

Plasma fibrinogen is linked with thrombosis. Fibrates were reported to decrease fibrinogen levels by inhibiting its expression via PPARα activation [45]. Since fibrinogen level predicted mortality in the BIP Study [46], bezafibrate-mediated reduction of fibrinogen may be one of the factors predicting mortality. Pemafibrate reduces fibrinogen more than fenofibrate [47]. 

## 3. Applications of Pemafibrate for Dyslipidemia

### 3.1. Pemafibrate Improves Lipid, Lipoprotein, and Apolipoprotein Metabolism

Kowa launched pemafibrate in June 2018 in Japan. JAS has classified pemafibrate into a novel category of drug therapy for dyslipidemia “SPPARMα”, which is completely different from fibrates [48]. Effects of pemafibrate on dyslipidemia from both a basic and clinical point of view are summarized in Appendix A. 

Pemafibrate may have a better benefit-risk balance superior to fibrates. IAS and the R3i Foundation have published a consensus statement on the concept of SPPARMα [49]. Clinical trials on pemafibrate have been performed mainly in Japan, as summarized below.

Chylomicron remnants and VLDL remnants (IDL) are called “remnant lipoproteins” and are proatherogenic [50,51]. Small dense LDL particles are also proatherogenic. Hypertriglyceridemia is often associated with increased small dense LDL and decreased HDL-C. Pemafibrate was demonstrated to significantly decrease remnant lipoprotein cholesterol (RemL-C), non-HDL-C, and levels of apo B, apo B-48, and apo C-III. By GP-HPLC analysis, pemafibrate was shown to dose-dependently reduce cholesterol concentrations of small LDL and increase those of small HDL [47]. A meta-analysis of pemafibrate, in comparison with fenofibrate, has established its efficacy and safety in patients with dyslipidemia [52]. The effects of pemafibrate on the reduction of serum TG and non-HDL-C levels and the increase in HDL-C were comparable to those for fenofibrate.

Postprandial hypertriglyceridemia is markedly atherogenic because atherogenic chylomicron remnants are increased in this condition. Pemafibrate attenuated postprandial hypertriglyceridemia in patients with dyslipidemia, while both fasting and non-fasting levels of serum TG, RemL-C, and apo B-48 were reduced [53]. Pemafibrate administration significantly reduced the AUC for postprandial TG level. Similar results were reported in pemafibrate-treated diabetic patients [54].

The particle number of each lipoprotein subclass can also be evaluated by GP-HPLC [55]. Pemafibrate reduced the particle number of atherogenic small LDL, while it increased that of small HDL, which is assumed to be antiatherogenic [56]. The effect of pemafibrate on the cholesterol efflux capacity of HDL from macrophages was investigated in patients with dyslipidemia [53]. HDL-C, HDL_3_-C, preβ1HDL, and apo A-1 levels were significantly increased by pemafibrate. The HDL from pemafibrate-treated patients possessed a significantly greater cholesterol efflux capacity compared with that from the placebo-treated patients. It increased the levels of FGF21, which may increase the expression of ABCA1 and ABCG1 involved in cholesterol efflux [35,37,38,39,53,57].

### 3.2. Pemafibrate in Combination with Statins

Combined treatment of fibrates in addition to statins increased the risk of rhabdomyolysis, particularly in CKD patients. The interaction of pemafibrate with high-dose statins was evaluated in healthy male volunteers [58]. The coadministration of pemafibrate with various statins (pravastatin, simvastatin, fluvastatin, atorvastatin, pitavastatin, or rosuvastatin) did not increase AUC or Cmax of pemafibrate or statins, excluding the possibility of drug–drug interactions between pemafibrate and statins. 

Hypertriglyceridemic patients taking pitavastatin were given pemafibrate for 12 weeks [59]. The fasting TG levels were reduced by 46.1% in the 0.1 mg/day pemafibrate group, 53.4% in the 0.2 mg/day group, and 52.0% in the 0.4 mg/day group, whereas it was decreased by 6.9% in the placebo group. Moreover, hypertriglyceridemic patients taking statins were treated for 24 weeks with pemafibrate (0.2–0.4 mg/day), causing the serum TG level to consistently reduce without any significant increases in adverse effects. Coadministration of pemafibrate with statins resulted in an improvement in the liver function test values. Pemafibrate treatment slightly increased serum creatinine and decreased eGFR, although these changes were clinically negligible.

### 3.3. Pemafibrate for Patients with CKD

CKD patients, even under hemodialysis, do not show high LDL-C levels, except for those with nephrotic syndrome. They are usually accompanied by an increase in TG and a decrease in HDL-C [60]. In trials of patients with CKD, such as 4D [61] and AURORA [62], statins did not prove to lower CV events. The SHARP study [63] demonstrated that intestinal cholesterol transporter inhibitor ezetimibe significantly reduces CV events. 

Typical lipoprotein abnormalities associated with CKD patients include hypertriglyceridemia with increased remnants and small dense LDL as well as a reduction in HDL-C. However, treatment of dyslipidemic patients with CKD was difficult because fibrates, except for clinofibrate, were metabolized and excreted from the kidneys. In humans, the liver mainly metabolizes pemafibrate, which is excreted from the liver into bile, with only 14.5% excretion into urine [64]. Its metabolites in plasma are mainly oxidized form at the benzyl position and a mixture of glucuronide conjugate of dicarboxylated form and *N*-dealkylated form (Figure 1) [65]. Less than 0.5% of the unmetabolized pemafibrate is excreted into urine and most of the metabolized compounds excreted into urine do not have PPARα agonist activity.

The blood concentrations of fibrates, such as clofibrate, gemfibrozil, fenofibrate, and bezafibrate, which are mainly metabolized in the kidney, are increased in CKD patients. In contrast, serum concentrations of pemafibrate were not elevated in patients with severe renal dysfunction [66]. Pemafibrate administration for long term was shown effective and safe in dyslipidemic patients, including those with renal dysfunction [67]. Blood pemafibrate concentration was not increased, even after administering repeated dosages. A recent pharmacokinetic study (PALT02) in patients with a marked renal dysfunction demonstrated that the blood concentrations of pemafibrate were not significantly increased, even in those receiving hemodialysis [68]. Considering its metabolic route and pharmacokinetic data, we can safely administer pemafibrate, even in patients with CKD. Compared with fibrates, pemafibrate may have a better benefit–risk balance, and its administration may be beneficial to patients for whom the use of conventional fibrates is limited.

### 3.4. Characteristic Features of Pemafibrate in Comparison with Fibrates

Compared with fenofibrate, pemafibrate showed better efficacy in the first three clinical trials [47,69,70]. TG-lowering effect of pemafibrate 0.4 mg/day (0.2 mg BID) was stronger than for fenofibrate 100 mg/day and was comparable to fenofibrate 200 mg/day. The increment of serum HDL-C by pemafibrate is usually larger than for fenofibrate [47]. Incidence of adverse events was not significantly different between patients treated with pemafibrate and the placebo, while it was markedly lower in pemafibrate-treated patients than in fenofibrate-treated patients. Fibrates were often reported to worsen the values of the renal function test, such as serum levels of creatinine and cystatin C, and eGFR [71,72,73]. Kidney function-related adverse events were markedly rare in pemafibrate-treated patients. In contrast, fenofibrate increased serum creatinine and cystatin C levels and decreased eGFR levels. Characteristic features of SPPARMα and pemafibrate are summarized in Table 1.

Fibrates often increased the values of the liver function test, such as ALT and γ-GT, and its mechanism was attributed to the activation of PPARα [74]. However, pemafibrate treatment rather decreased these values. Differences between pemafibrate and fenofibrate were remarkable, with the regard to liver function tests and levels of serum creatinine and FGF21. Especially, pemafibrate reduced ALT, γ-GT, and ALP levels by approximately 8 U/L, 24 U/L, and 70–80 U/L, respectively. Pemafibrate-mediated changes in liver and kidney function test values may be the most striking characteristic features of pemafibrate.

Pemafibrate enhanced the expression of cholesterol efflux-related genes in macrophages such as ABCA1 and ABCG1 and attenuated that of proinflammatory genes, including VCAM1, F4/80 (macrophages), and IL-6. The effects on basic parameters were markedly different between pemafibrate and fenofibrate [34].

## 4. Pemafibrate Affects Glucose Metabolism and Insulin Resistance

Twenty-four-week treatment with pemafibrate in type 2 diabetic patients with hypertriglyceridemia demonstrated effects on lipids and lipoproteins similar to those reported in phase 2 and 3 studies. It is noteworthy that pemafibrate significantly decreased fasting levels of blood glucose and insulin in comparison to the placebo [54]. In the PROVIDE study, 52-week treatment with pemafibrate in type 2 diabetic patients with hypertriglyceridemia [75] markedly decreased serum TG and non-HDL-C levels, and increased HDL-C levels.

A post-hoc analysis of six phase 2 and phase 3 randomized double-blind placebo-controlled trials in Japan evaluated the effects of pemafibrate on glucose metabolism markers in 1253 patients, randomized to placebo or pemafibrate 0.1 mg/day, 0.2 mg/day, or 0.4 mg/day [76]. Pemafibrate significantly decreased fasting glucose, insulin, and HOMA-IR compared to the placebo, with the greatest decrease observed in pemafibrate 0.4 mg/day. These results indicated that pemafibrate may ameliorate insulin resistance.

Using a hyperinsulinemic-euglycemic clamp to evaluate liver or peripheral insulin resistance [77], pemafibrate was shown to significantly increase the hepatic glucose uptake rate, suggesting that it ameliorates insulin resistance. Pemafibrate was demonstrated to attenuate high-fat diet-induced weight gain and decrease plasma glucose and insulin levels in diet-induced obesity mice, by increasing plasma FGF21 [44]. Pemafibrate enhanced the expression of genes involved in thermogenesis and fatty acid β-oxidation and improved obesity-induced metabolic abnormalities. 

ABCA1 plays a crucial role in cholesterol and phospholipids efflux from cells to HDL. A deficiency of ABCA1 causes Tangier disease characterized by a marked reduction in HDL-C, orange tonsils, hepatosplenomegaly, and enhanced atherosclerosis [78]. The oral glucose tolerance test in patients with Tangier disease indicated a progressive increase in plasma glucose, yet not insulin concentration, suggesting a lower insulinogenic index in patients than in the non-diabetic controls [79]. Since pancreatic β-cells express ABCA1, glucose-stimulated insulin secretion may be impaired in Tangier disease patients. Pancreatic ABCA1 may play a role in β cell cholesterol homeostasis, thus, affecting insulin secretion [80]. Pemafibrate increased ABCA1 mRNA and protein levels, decreased cellular cholesterol content in INS-1 cells, and enhanced insulin secretion by regulation of ABCA1 expression in β cells [81]. Furthermore, it improved HOMA-IR, suggesting that it may attenuate insulin resistance in a meta-analysis [52]. Figure 2 illustrates the possible main targets of pemafibrate in conditions of dyslipidemia resulting from visceral obesity, metabolic syndrome, and NAFLD/NASH.

## 5. Pemafibrate for NAFLD, NASH, and PBC

### 5.1. NAFLD

Recently, NAFLD and NASH are important because they are associated with enhanced ASCVD. PPARα-null mice showed advanced fatty liver and steatohepatitis [82], while patients with NASH demonstrated a reduction in hepatic PPARα expression [83]. PPARα agonists can be applicable for NAFLD. However, unsuccessful CV event outcomes in fibrates were attributed partly to adverse reactions, such as liver and renal dysfunction. In contrast, clinical trials on pemafibrate had consistently decreased levels of serum ALT, ALP, γ-GT, and total bilirubin. More prominent effects were observed in patients with higher liver function values than in those with normal values [76], suggesting a possible application of pemafibrate for patients with NAFLD or NASH.

The effects of pemafibrate on NAFLD/NASH from a basic and clinical point of view are summarized in Appendix A In the NAFLD/NASH mouse model, pemafibrate reduced liver function test values and improved fatty liver, ballooning, inflammation, and fibrosis [84,85,86,87]. Regarding the mechanisms through which pemafibrate ameliorates NAFLD, the genes for β-oxidation in, and lipid transport out of the liver are enhanced and the energy metabolism is also upregulated via the induction of the UCP3 gene. Kanno et al. [87] evaluated the effect of pemafibrate on hepatic steatosis in a novel mouse model of diet-induced steatohepatitis-related cardiomyopathy fed a high-fat, high-cholesterol, high-sucrose, and bile acid diet (NASH diet). Mice were fed an 8-week NASH diet with or without pemafibrate (0.1 mg/kg). More macrophage infiltration and fibrosis were demonstrated in the livers of the NASH diet group compared to the control diet group. Steatohepatitis with increased free cholesterol content and cholesterol crystals was established. Free cholesterol was also accumulated in the heart accompanied by concentric hypertrophy and impaired left ventricular ejection fraction. Enhancement of the NOD-like receptor and PI3 kinase-Akt pathways was observed. The mRNA and protein expression of inflammasome-related genes, including caspase-1, NLRP3, and IL-1β were upregulated in the liver and heart. Thus, these data demonstrated that pemafibrate attenuated hepatic steatosis, steatohepatitis, and cardiac dysfunction. Pemafibrate can recover hepatic fibrosis and cardiac dysfunction, even after the development of steatohepatitis-related cardiomyopathy.

To examine possible clinical applications of pemafibrate in patients with NAFLD, several single-arm preliminary studies, in Japan, retrospectively evaluated the efficacy of pemafibrate in a small number of patients with NAFLD [88,89,90,91,92,93]. These studies consistently showed that pemafibrate improved serum levels of ALT, ALP, and γ-GT as well as fibrosis markers such as AST/platelet ratio index and FIB-4 index, although there were some variations. A prospective single-arm study also evaluated the efficacy of pemafibrate in a small number of 20 NAFLD patients with dyslipidemia, who were administered pemafibrate (0.1 mg BID) for 12 weeks [94]. The change in serum ALT levels from the baseline to week 12 was set as a primary endpoint. Serum ALT levels were significantly decreased from the baseline to week 12. Serum TG, HDL-C, total fatty acid, saturated fatty acid, and unsaturated fatty acid levels were significantly improved.

A double-blind, placebo-controlled, randomized multicenter, phase 2 trial in Japan has been reported recently [95]. A total of 118 patients were randomized to either 0.2 mg pemafibrate or placebo, twice daily, and treated for 72 weeks. As inclusion criteria, the following patients were enrolled: liver fat content of ≥10% by MRI-PDFF; liver stiffness of ≥2.5 kPa by MRE; elevated ALT levels. The percentage change in MRI-PDFF from the baseline to week 24 was set as the primary endpoint. MRE-based liver stiffness, ALT, serum liver fibrosis markers, and lipid parameters were the secondary endpoints. No significant difference was observed between the groups in the primary endpoint, however, MRE-based liver stiffness was significantly reduced by pemafibrate at week 48 compared to placebo and was maintained until week 72. Significant reductions in ALT and LDL-C were also demonstrated. Thus, pemafibrate showed a significant reduction in MRE-based liver stiffness, yet not liver fat content, suggesting that it may be a promising new therapeutic agent for NAFLD/NASH and a candidate for combination therapy with agents that may reduce liver fat content. Another trial is in progress to evaluate the efficacy and safety of a combination of pemafibrate-extended-release (ER) and SGLT-2 inhibitor, tofogliflozin, in patients with NASH and liver fibrosis (NCT05327127) [96].

### 5.2. PBC

Fenofibrate [97] and bezafibrate [98] lowered liver function test values in PBC patients. Pemafibrate demonstrated favorable effects on liver function test values in patients with dyslipidemia, suggesting that it may improve liver function in PBC patients. To prove the safety issues, a pharmacokinetic study of pemafibrate is currently ongoing on patients with PBC (JapicCTI-173728). In a recent pilot study, pemafibrate was shown to improve liver function tests in a small number of patients with PBC [99]. It may be crucial to explore the effects of pemafibrate for patients with PBC in a large-scale multicenter trial.

## 6. Effects of Pemafibrate on Endothelial Function, Neointimal Formation, Inflammation, and Atherosclerosis

Effects of pemafibrate on endothelial function, neointimal formation, inflammation, and atherosclerosis from a basic and clinical point of view are summarized in Appendix A.

### 6.1. Endothelial Function

The effect of pemafibrate on revascularization was evaluated in a mouse model of hindlimb ischemia [100]. It enhanced blood flow recovery and capillary formation in ischemic limbs, and phosphorylation of eNOS was increased. Cultured endothelial cells treated with pemafibrate increased the formation of the network and migratory activity, which was attenuated by a NOS inhibitor. Pemafibrate increased plasma levels of FGF21. Adenovirus-mediated FGF21overexpression increased blood flow recovery, the density of capillaries, and phosphorylation of eNOS in ischemic limbs in mice. Endothelial cell network formation and migration in cultured endothelial cells were enhanced by treatment with the FGF21 protein, which was canceled by NOS inhibitor pretreatment or in eNOS knockout mice. Revascularization in response to ischemia by pemafibrate may be induced partly via direct and FGF21-mediated modulation of endothelial cell function. Yoshida et al. [101] investigated the effect of the combined administration of pitavastatin and pemafibrate on endothelial dysfunction in a Dahl rat model of salt-sensitive hypertension and insulin resistance. A combination of high-dose pitavastatin and pemafibrate significantly increased acetylcholine-induced endothelial relaxation rates compared to the vehicle, suggesting that endothelial dysfunction is ameliorated. Endothelium-dependent vascular responses to acetylcholine were also evaluated in streptozotocin-diabetic mice [102]. Three-week treatment with pemafibrate reduced serum TG and non-HDL-C, while it also decreased the levels of arachidonic acid, thromboxane B2, prostaglandin E2, leukotriene B4, and 5-hydroxyeicosatetraenoic acid, which were increased by diabetic state. Pemafibrate also decreased palmitic acid and stearic acid levels. Condition of diabetes-induced endothelial dysfunction, which was ameliorated by pemafibrate, via a reduction in vasoconstrictive eicosanoids and free fatty acids levels.

### 6.2. Neointimal Formation

Konishi et al. [103] reported the effects of pemafibrate on vascular response to balloon injury in LDL receptor-null pigs. The animals were fed a cholesterol-rich diet, allocated randomly to pemafibrate and control groups, and the balloon injury was created 2 weeks after drug administration. The average ratio of macrophages to plaque area, yet not intimal area, was significantly reduced in pemafibrate group compared to the control group. The mRNA expressions of C-Jun, NFκB, and MMP9 were significantly reduced by pemafibrate, suggesting that it may inhibit inflammatory responses after balloon injury. Similarly, pemafibrate inhibited neointima formation after vascular injury in mice fed normal chow and high-fat diet [104]. Pemafibrate increased serum concentrations of FGF21 and decreased those of insulin in high-fat diet mice. Pemafibrate, yet not bezafibrate, attenuated the proliferation and DNA synthesis of VSMCs. This effect of pemafibrate was abolished by PPARα knockdown, suggesting that it attenuates neointima formation after vascular injury by inhibiting VSMC proliferation via PPARα.

Iwata et al. [105] examined the effect of pemafibrate on coronary stent-induced arterial inflammation and neointimal hyperplasia in Yorkshire pigs. Intracoronary molecular-structural near-infrared fluorescence and optical coherence tomography imaging demonstrated that pemafibrate reduced coronary stent-induced cellular inflammation and neointimal hyperplasia. It also suppressed the expression of TNF-α and MMP-9 in the neointima and increased smooth muscle cell differentiation markers, calponin, and smoothelin, via STAT3-myocardin axes. Taken together, pemafibrate can be a novel strategy to prevent stent restenosis.

### 6.3. Inflammation and Atherosclerosis

Several animal studies have shown preventive effects of pemafibrate on inflammation and atherosclerosis. It decreased mRNA expressions of small intestine apo B and liver apo C-III in humans and apo E2 knock-in mice fed a high-fat/high-cholesterol diet [37]. Pemafibrate (1.0 mg/kg) reduced atherosclerotic lesions better than fenofibrate (250 mg/kg). Anti-inflammatory effects of pemafibrate were demonstrated by significantly reduced mRNA expressions of F4/80, VCAM1, and IL-6 in atherosclerotic lesions.

The effects of pemafibrate on proteomics and high-dimensional clustering on vein graft tissues were evaluated to explore the seeds for preventing vein graft failure [106]. In vivo mice experiments using small interfering RNA of macrophage-targeted PPARα and pemafibrate showed that the inhibition of PPARα accelerates the development and inflammation of vein graft lesions. Proteomic analysis of vein grafts showed changes in proteome related to lipid and fatty acid metabolism regulated by the PPARs, immune responses, matrix remodeling, and hematopoietic cell mobilization. Pemafibrate-mediated PPARα activation inhibited the development and inflammation of vein graft lesions, while gene silencing of PPARα worsened plaque formation.

Pemafibrate reversed the enhanced neutrophil adhesion on the atheroprone femoral artery of high-fat diet-fed LDL receptor knockout mice caused by histone H3 citrullination [107]. It also prevented AAA rupture in apo E-null mice induced by subcutaneous infusion of angiotensin II [108]. This pemafibrate effect was associated with a reduction in reactive oxygen species, extracellular matrix degradation, and inflammation in the aortic wall. The protective effect of pemafibrate against AAA rupture was partly due to the anti-oxidative effect of the activated catalase in smooth muscle cells.

## 7. Effects of Pemafibrate on Events of ASCVD

A multinational, double-blind, randomized, controlled trial, PROMINENT study [109] was performed worldwide to investigate the effect of pemafibrate on CV events and the results have recently been reported [110]. Type 2 diabetic patients with hypertriglyceridemia (TG: 200–499 mg/dL) and low HDL-C (≤40 mg/dL) were assigned to receive pemafibrate (0.2 mg tablets twice daily) or a placebo. This study enrolled nearly 10,000 patients (66.9% with CV disease) receiving guideline-directed lipid-lowering therapy, or those who could not receive statin without adverse effects alongside LDL-C levels less than 100 mg/dL. A composite of nonfatal myocardial infarction, ischemic stroke, coronary revascularization, or CV deaths was set as the primary endpoint. Changes in lipid levels at 4 months by pemafibrate compared to the placebo were −26.2% for TG, −25.8% for VLDL-cholesterol, −25.6% for remnant cholesterol, −27.6% for apo C-III, and 4.8% for apo B. No significant difference was noted in the primary endpoint between the pemafibrate and placebo groups. The incidence of serious adverse events was not significantly different between the groups, although the use of pemafibrate was associated with a higher incidence of adverse renal events and venous thromboembolism and a lower incidence of NAFLD. Regarding adverse renal events, the very mild increase in serum creatinine and a little decrease in eGFR during the pemafibrate treatment were completely recovered after stopping treatment, suggesting that creatinine synthesis may be increased by PPARα activation. Therefore, the observed changes in renal function parameters do not mean renal dysfunction. In this trial, a mild increase in apo B and apo B-containing lipoproteins concentrations as well as LDL-C may be attributed to the outcomes [111]. However, in contrast to the previously reported data in Japan [54,112], levels of TG, VLDL-cholesterol, and remnant cholesterol were also decreased in the placebo group, resulting in milder treatment effects of pemafibrate. An increase in HDL-C level was also mild in the pemafibrate group. It slightly increased apo B and LDL-C levels, suggesting a small increase in the number of apo B-containing lipoproteins. Since the dose of statins is usually much higher in Western countries than in Japan, the presence of moderate to high-intensity statins may have negated the apo B-lowering effects of pemafibrate [111].

Recently, the STRENGTH trial, which analyzed a combined formula of eicosapentaenoic acid and docosahexaenoic acid, did not demonstrate a significant decrease in apo B levels as well as the incidence of CV events [113]. All of the patients had already been treated with statins (about half were high-intensity statins). Recently, it has become very difficult to show the benefits of a test drug in patients treated with high-intensity statins. Furthermore, the mean body mass indexes of the pemafibrate and placebo groups were nearly 32, which was much larger than in Japanese patients. It may be possible that the dose of pemafibrate 0.4 mg/day was not enough to achieve a larger TG reduction, as reported in Japanese patients. In the meta-regression analysis of randomized controlled trials, a reduction in the serum TG level was accompanied by a lower major vascular event risk after adjusting the LDL-C level [114]. Taken together, the negative results of the PROMINENT study [110] may not negate the importance of TG-lowering therapy for the prevention of ASCVD events and pancreatitis. 

## 8. Conclusions

Pemafibrate is the first SPPARMα based upon a novel concept, which has potent and high selectivity for PPARα and is distinctly different from fibrates. It is not metabolized by the kidney, yet is mainly by the liver and is secreted into the bile. Thus, it can be used in patients with CKD. It does not have significant interactions with statins and its coadministration with any statin is safe. Pemafibrate may be administered in patients with a variety of metabolic diseases. The possible applications for metabolic diseases and conditions are illustrated in Figure 3. 

Pemafibrate may have a better risk-benefit balance than conventional fibrates and is a safe drug to use for patients taking statins and those with CKD or NAFLD. Pemafibrate, the first SPPARMα, is already marketed in Japan, ahead of the rest of the world, and is expected to demonstrate better efficiency than fibrates, thereby providing a novel therapeutic option for dyslipidemia as well as NAFLD and NASH.

## Figures and Tables

**Figure 1 metabolites-13-00626-f001:**
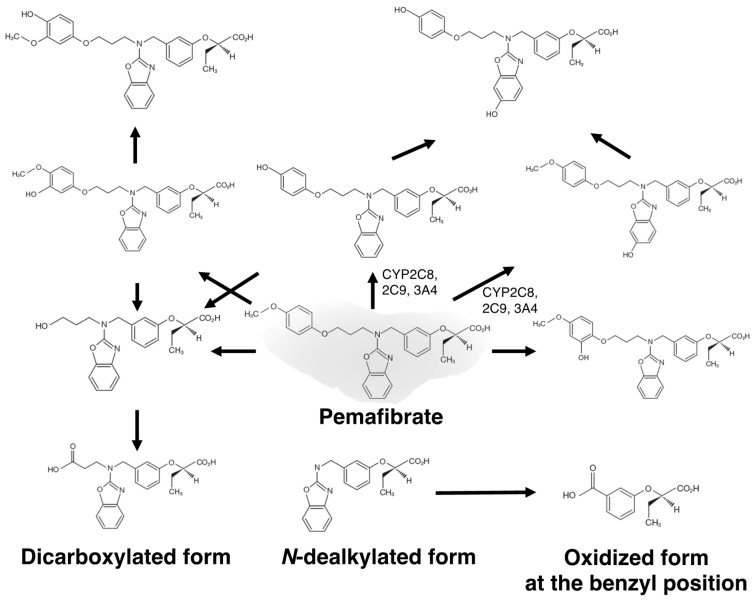
Estimated metabolic pathways of pemafibrate.

**Figure 2 metabolites-13-00626-f002:**
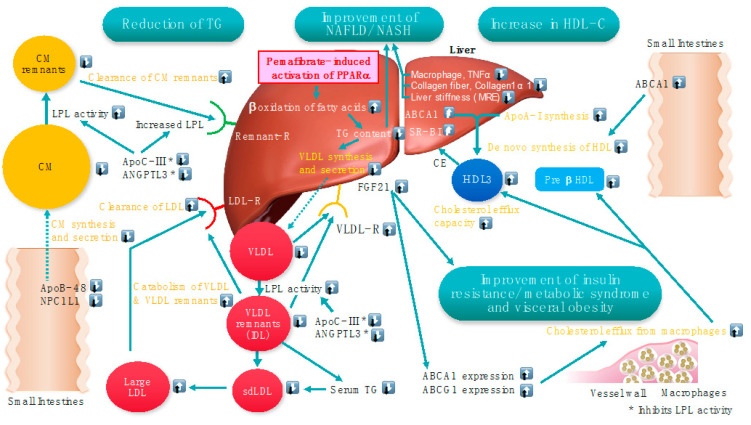
Possible main targets of pemafibrate in conditions of dyslipidemia resulting from visceral obesity, metabolic syndrome, and NAFLD/NASH. Modified from a Figure in Reference [34]. * Both ApoC-III and ANGPTL3 inhibit LPL activity. ⍐: Increase ⍗: Decrease.

**Figure 3 metabolites-13-00626-f003:**
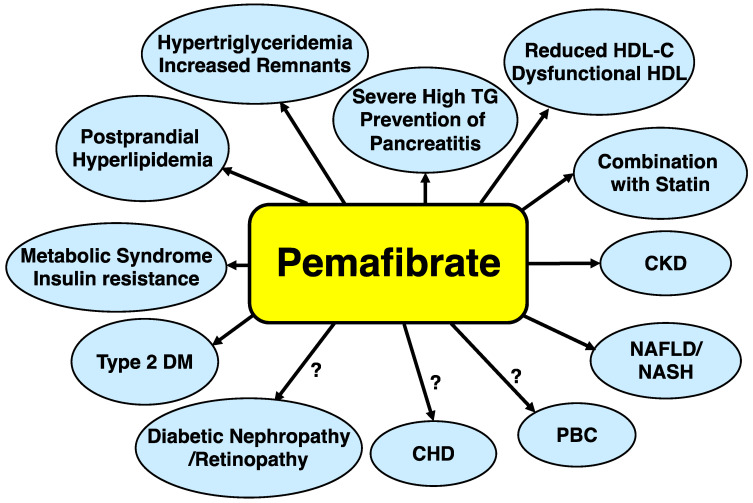
Possible applications of pemafibrate for metabolic diseases. Abbreviations: CKD: chronic kidney disease; CHD: coronary heart disease; HDL-C: high-density lipoprotein cholesterol; NASH: nonalcoholic steatohepatitis; PBC: primary biliary cholangitis; RemL-C: remnant lipoprotein cholesterol; TG: triglyceride.

**Table 1 metabolites-13-00626-t001:** Characteristics Features of SPPARMα, Pemafibrate.

■ Is mainly metabolized in the liver, not affected by renal function
■ Shows little influence on renal function
■ Can be used with statins as it has no known drug-drug interactions with statins
■ Increases LPL, but decreases apoC-III, thereby activating LPL
■ Decreases TG, remnant lipoproteins and small dense LDL
■ Improves postprandial hyperlipidemia
■ Increases HDL-C (especially small spherical HDL and preb HDL) and activates anti-atherosclerotic HDL function
■ Enhances β-oxidation in the liver
■ Decreases fibrinogen
■ Favorably affects liver function and may improve NASH/NAFLD
■ Improves insulin resistance and glucose metabolism

Abbreviations: LPL, lipoprotein lipase; apo, apolipoprotein; TG, triglyceride; LDL, low-density lipoprotein; HDL-C, high-density lipoprotein cholesterol; NAFLD, non-alcoholic fatty liver disease; NASH, non-alcoholic steatohepatitis.

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
