# Peer review of "Novel Selective PPARα Modulator Pemafibrate for Dyslipidemia, Nonalcoholic Fatty Liver Disease (NAFLD), and Atherosclerosis [Author-notes fn1-metabolites-13-00626]"

_metabolites, 2023, doi:10.3390/metabo13050626_

Round 1

Reviewer 1 Report

The review is quite extensive but very descriptive.  without schematics for each headline study, it is difficult to follow.  Even though this review appears like an advertisement for SPPARMa, it contains significant amount of information but hard to follow without schematics and tables describing studies.  This exercise of providing schematics will also make the authors to think of ways of making the studies clear to readers.  some specific suggestions are as follows: 130-131--what is mechanisms of predicting mortality?  change it to "factors predicting mortality" 

134 heading---apoporptein  change it to apolipoprotein

196-197  what are benzoyl oxidized oxidants?  In fact, here, I suggest that the authors create a diagram showing the structures of metabolites of drug metabolism to make it clear to the readers.   

280-281  ALT, g-GT were already defined and here is it redefined indicating lack of careful editing

Author Response

Responses to Reviewer 1:

Thank you very much for your valuable comments. We have added the following 3 new Tables for readers to easily understand the contents.

Table 1. Effects of Pemafibrate on Dyslipidemia.

Table 3. Effects of Pemafibrate on NAFLD/NASH.

Table 4. Effects of Pemafibrate on Atherosclerosis and ASCVD.

Accordingly, Table 1 (Characteristic Features of SPPARMα, Pemafibrate) was renamed as Table 2.

The other parts were revised as you suggested. We have added the section of abbreviations used in this review.

Regarding the metabolites section, we have changed the sentence as follows:

“Its metabolites in plasma are mainly oxidized benzyl position of pemafibrate and a mixture of glucuronide conjugate and N-dealkylated pemafibrate (new Figure 1) [63].”

We have added new Figure 1 as a diagram showing the structures of drug metabolites.

Following the suggestion of Reviewer 3, we have also added new Figure 2 which illustrates the possible main targets of pemafibrate in conditions of dyslipidemia resulting from visceral obesity, metabolic syndrome and NAFLD/NASH. Accordingly, previous Figure 1 was renamed as new Figure 3.

Finally, the editorial office suggested there were some overlaps with our previous two reviews. We added the following sentence in the title page and rephrased these parts.

*The current paper is an extended and updated version of the following two reviews:

1) Yamashita S, Masuda D, Matsuzawa Y: Clinical applications of a novel selective PPARα modulator, pemafibrate, in dyslipidemia and metabolic diseases. J Atheroscler Thromb 2019;26(5):389-402.

2) Yamashita S, Masuda D, Matsuzawa Y: Pemafibrate, a new selective PPARα modulator: drug concept and its clinical applications for dyslipidemia and metabolic diseases. Curr Atheroscler Rep 2020;22:5.

I hope the manuscript has been fully revised according to the reviewers’ suggestions.

Reviewer 2 Report

This review entitled "Novel Selective PPARalpha Modulator Pemafibrate for Dyslipidemia, Nonalcoholic Fatty Liver Disease (NAFLD) and Atherosclerosis" is well-written and very important because Pemafibrate as novel Selective PPARalpha Modulator in obesity, metabolic syndrome, NAFLD and atherosclerosis. The authors summarized the recent findings o pemafibrate. Pemafibrate is a drug that has shown better effects on reducing serum triglycerides and elevating HDL-C compared to fenofibrate. It also improves liver function test values and has little effect on serum creatinine levels and estimated glomerular filtration rate. Pemafibrate has minimal drug-drug interactions with statins and can be used safely in patients with chronic kidney disease. This review is very important considering that both the pathogenesis and the therapeutic strategy of NAFLD are still unclear.

Minor Comment:

I am of the opinion that a figure should be added summarizing the main targets on which this drug acts in conditions of dyslipidemia resulting from obesity, metabolic syndrome, and NAFLD/NASH.

Author Response

Responses to Reviewer 2:

Thank you very much for your valuable and very supportive comments.

As you suggested, we have added new Figure 2 which illustrates the possible main targets of pemafibrate in conditions of dyslipidemia resulting from visceral obesity, metabolic syndrome and NAFLD/NASH. Accordingly, previous Figure 2 was renamed as new Figure 3.

Furthermore, according to the suggestions of Reviewer 1, we have added the following 3 new Tables for readers to easily understand the contents.

Table 1. Effects of Pemafibrate on Dyslipidemia.

Table 3. Effects of Pemafibrate on NAFLD/NASH.

Table 4. Effects of Pemafibrate on Atherosclerosis and ASCVD.

Accordingly, Table 1 (Characteristic Features of SPPARMα, Pemafibrate) was renamed as Table 2.

We have added the section of abbreviations used in this review.

Following the suggestions of Reviewer 1 on the metabolites of pemafibrate, we have changed the sentence as follows:

“Its metabolites in plasma are mainly oxidized benzyl position of pemafibrate and a mixture of glucuronide conjugate and N-dealkylated pemafibrate (new Figure 1) [63].”

We have added new Figure 1 as a diagram showing the structures of drug metabolites.

Finally, the editorial office suggested there were some overlaps with our previous two reviews. We added the following sentence in the title page and rephrased these parts.

Following the suggestion of Reviewer 3, we have also added new Figure 2 which illustrates the possible main targets of pemafibrate in conditions of dyslipidemia resulting from visceral obesity, metabolic syndrome and NAFLD/NASH. Accordingly, previous Figure 1 was renamed as new Figure 3.

*The current paper is an extended and updated version of the following two reviews:

1) Yamashita S, Masuda D, Matsuzawa Y: Clinical applications of a novel selective PPARα modulator, pemafibrate, in dyslipidemia and metabolic diseases. J Atheroscler Thromb 2019;26(5):389-402.

2) Yamashita S, Masuda D, Matsuzawa Y: Pemafibrate, a new selective PPARα modulator: drug concept and its clinical applications for dyslipidemia and metabolic diseases. Curr Atheroscler Rep 2020;22:5.

I hope the manuscript has been fully revised according to the reviewers’ suggestions.

Reviewer 3 Report

Thanks for the opportunity to review the manuscript entitled ”Novel Selective PPARα Modulator Pemafibrate for Dyslipidemia, Nonalcoholic Fatty Liver Disease (NAFLD) and Atherosclerosis”.

The article is very well documented and contains a lot of informations regarding the effects of the selective PPARα agonist, Pemafibrate. Its actions are compared with those of other fibrates, and also in combination with statins.

I suggest the authours to present some data in tables, to be easier for readers to follow all these informations, as the quantity of data is high and complex. So to add some tables (I suggest 3), which should present in summary the mechanisms by which Pemafibrate acts in dyslipidemia, in non-alcoholic fatty liver disease/non-alcoholic steatohepatitis and also in atherosclerosis/ASCVD.

Author Response

Responses to Reviewer 3:

Thank you very much for your valuable and very supportive comments.

As you and Reviewer 1 suggested, we have added the following 3 new Tables for readers to easily understand the contents.

Table 1. Effects of Pemafibrate on Dyslipidemia.

Table 3. Effects of Pemafibrate on NAFLD/NASH.

Table 4. Effects of Pemafibrate on Atherosclerosis and ASCVD.

Accordingly, Table 1 (Characteristic Features of SPPARMα, Pemafibrate) was renamed as Table 2.

We also have added new Figure 2 which illustrates the possible main targets of pemafibrate in conditions of dyslipidemia resulting from visceral obesity, metabolic syndrome and NAFLD/NASH. Accordingly, previous Figure 2 was renamed as new Figure 3.

We also added the section of abbreviations used in this review.

Following the suggestions of Reviewer 1 on the metabolites of pemafibrate, we have changed the sentence as follows:

“Its metabolites in plasma are mainly oxidized benzyl position of pemafibrate and a mixture of glucuronide conjugate and N-dealkylated pemafibrate (new Figure 1) [63].”

We have added new Figure 1 as a diagram showing the structures of drug metabolites.

Finally, the editorial office suggested there were some overlaps with our previous two reviews. We added the following sentence in the title page and rephrased these parts.

*The current paper is an extended and updated version of the following two reviews:

1) Yamashita S, Masuda D, Matsuzawa Y: Clinical applications of a novel selective PPARα modulator, pemafibrate, in dyslipidemia and metabolic diseases. J Atheroscler Thromb 2019;26(5):389-402.

2) Yamashita S, Masuda D, Matsuzawa Y: Pemafibrate, a new selective PPARα modulator: drug concept and its clinical applications for dyslipidemia and metabolic diseases. Curr Atheroscler Rep 2020;22:5.

I hope the manuscript has been fully revised according to the reviewers’ suggestions.

Round 2

Reviewer 1 Report

authors have addressed previous concerns

Author Response

Thank you very much for your review!

Reviewer 2 Report

The authors added Figure 2. Possible Main Targets of Pemafibrate in Conditions of Dyslipidemia Resulting from Visceral Obesity, Metabolic Syndrome and NAFLD/NASH which is very illustrative and clear.

After the reviews of this manuscript, the authors accepted the remarks and suggestions and entered them into the new version of the article. So, in my opinion, this manuscript is suitable for publication in Metabolites.

Author Response

Many thanks for your review!

Reviewer 3 Report

I agree with the changes made by the authors.

Author Response

Thank you very much for your kindly review!